# Flavonol Glycosides from *Eugenia uniflora* Leaves and Their In Vitro Cytotoxicity, Antioxidant and Anti-Inflammatory Activities

**Ayodeji Oluwabunmi Oriola** [1,*] **, Gugulethu Mathews Miya** [1] **, Moganavelli Singh** [2] **and Adebola Omowunmi Oyedeji** [1]

1 Department of Chemical and Physical Sciences, Walter Sisulu University, Mthatha 5117, South Africa; gmiya@wsu.ac.za (G.M.M.); aoyedeji@wsu.ac.za (A.O.O.)

2 Nano-Gene and Drug Delivery Group, Discipline of Biochemistry, University of KwaZulu-Natal, Private Bag, Durban X54001, South Africa; singhm1@ukzn.ac.za

\* Correspondence: aoriola@wsu.ac.za; Tel.: +27-65-593-4742

**Abstract:** In view of the extensive use of *Eugenia uniflora* leaves for the management of tumours and other chronic inflammatory diseases in traditional medicine, an activity-guided fractionation of its leaf ethanolic extract led to the isolation of two flavonol glycosides. Cytotoxicity study was based on the 3-(4,5-dimethylthiazol-2-yl)-2,5-diphenyl-2H-tetrazolium bromide (MTT) viability assay against the non-tumourigenic human embryonic kidney (HEK-293) cells, and the cancerous liver (Hep-G2) and cervical (HeLa) cell lines. Antioxidant tests were carried out using 2,2-diphenyl-1-picrylhydrazyl (DPPH), nitric oxide (NO) and hydrogen peroxide ($H_2O_2$) radical scavenging assays, while an in vitro anti-inflammatory test was conducted using egg albumin denaturation (EAD) assay. Based on comprehensive spectroscopic and spectrometric evidence, the compounds were elucidated as myricitrin (**1**) and a newly described compound, 5,7-dihydroxy-3-(3,4,5-trihydroxy-6-methyltetrahydropyran-2-yloxy)-2-(2,4,5-trihydroxyphenyl)chromen-4-one, named "unifloratrin (**2**)". The cytotoxicity of myricitrin (**1**) was comparable to 5-fluorouracil (standard drug), with a $CC_{50}$ of $8.5 \pm 2.2$ μg/100 μL against HeLa cells. It also demonstrated better antioxidant activity, with an $IC_{50}$ of $6.23 \pm 1.09$, $22.01 \pm 2.59$ and $30.46 \pm 1.79$ μM against DPPH, NO and $H_2O_2$ free radicals, respectively. At 20 μg/mL and an incubation time of 2 h, myricitrin was comparable to diclofenac (standard drug) in anti-inflammatory activity. This report may serve as a justification for the ethnomedicinal use of *E. uniflora*, while flavonol glycosides, such as myricitrin (**1**), could be further exploited as a candidate cytotoxic agent.

**Keywords:** *Eugenia uniflora*; flavonol glycosides; myricitrin; cytotoxicity; antioxidant; anti-inflammatory

## 1. Introduction

*Eugenia uniflora* L. (family Myrtaceae), commonly called "Suriname cherry" or "Pitanga cherry" is a small tree that is native to South America but is now widely distributed across other continents, especially in tropical African and Asian countries because of its ability to adapt to different habitats [1]. The leaf adaxial surface of *E. uniflora* is glabrous with ovate to elliptic leaf blade and acuminate to obtuse apex [2]. The plant grows up to 7 m high, producing white flowers and berry-like fruits that appear green when young and turn orange to dark red at maturity [3].

For many decades, the infusions or decoctions obtained from *E. uniflora* fresh and dried leaves have been used extensively in folk medicines as remedies for hypertension, fever, rheumatic pain, diarrhoea, inflammatory and stomach diseases caused by microbial infections, and for the management of chronic non-communicable diseases, including diabetes and cancer [4,5]. Biological studies on *E. uniflora* leaf extracts and essential oils have indicated their considerable antioxidant, antibacterial, antifungal, anti-inflammatory, wound healing and anticancer properties [3,5–9]. The leaf extract of *E. uniflora* has also

been implicated among a selected number of Nigerian medicinal plants to demonstrate considerable cytotoxicity against human prostate (DU-145), breast (JIMT-1) and pancreatic (BxPC-3) cancer cells [10].

*E. uniflora* is a popular aromatic plant in the family Myrtaceae having essential oil constituents, such as monoterpenes (*trans*-β-ocimene, *cis*-ocimene and β-pinene) and oxygenated- and non-oxygenated sesquiterpenes (germacrenes A, B, D, eugenilones A-N, seline-1,3,7-triene-8-one oxide and β-caryophyllene) [11–13]. A general phytochemical analysis of the leaves has revealed the presence of steroids, terpenes, tannins, phenolics and flavonoids [14]. Polyhydroxylindolizidine alkaloids, uniflorine A and B, have been isolated from the leaves of this plant as α-glucosidase inhibitors [15]. Flavonoid glycosides, such as kaempferol pentoside, myricetin galloyl hexoside, myricetin hexoside, myricetin pentoside, myricetin rhamnoside, quercetin galloyl hexoside, quercetin rhamnoside, quercetin hexoside and quercetin pentoside, have been identified in the Pitanga fruits, including the seeds [16,17]. The flavonoid-rich fraction of *E. uniflora* leaf was shown in another study to protect mice from murine sepsis, and by extension, showed antibacterial and anti-inflammatory properties [4].

Flavonoids are a group of bioactive secondary metabolites that are abundantly present in medicinal plants, including fruits and vegetables [18]. Structurally, they are made up of a benzo-γ-pyrone skeleton with different substituents to form seven sub-groups, namely flavanones, flavones, flavonols, flavanols, anthocyanins, isoflavones and flavanonols [19]. They are polyphenols synthesized by plants to act as defence mechanism against biotic and abiotic stresses [20]. Flavonoid-rich dietary products such as cherry, broccoli, tea, strawberries, parsley and chillies are known to help reduce the risks of cancer [19]. A more comprehensive phytochemical profiling of the ethyl acetate fraction of *E. uniflora* has been shown through a recent study to contain some flavonoids [21]. However, the said putative compounds are yet to be isolated, characterized, and potentiated.

Therefore, this study reports from the ethyl acetate fraction of *E. uniflora* leaves, the isolation, characterization, in vitro antioxidant, anti-inflammatory and cytotoxicity of two flavone glycosides, which are myricitrin (**1**) and 5,7-dihydroxy-3-(3,4,5-trihydroxy-6-methyltetrahydropyran-2-yloxy)-2-(2,4,5-trihydroxyphenyl)chromen-4-one, named unifloratrin (**2**).

## 2. Materials and Methods

### 2.1. General Experimental Procedures

The organic solvents used (Shalom Laboratory Supplies, Durban, KZN, South Africa) were of analytical grade and were redistilled before use. Thin-layer chromatography (TLC) was performed on pre-coated silica gel plates (Silica gel 60 F254, Merck KGaA, Darmstadt, Germany). The developed TLC plates were visualized under the ultraviolet (UV) light at 254 and 366 nm wavelengths and sprayed with 10% sulfuric acid (chromogenic reagent) for a general detection of organic compounds. Compound isolation was carried out via column chromatography (CC) on silica gel 60 adsorbent (0.063–0.200 mm, 70–230 mesh ASTM, Merck, Darmstadt, Germany). Absorbance values in the antioxidant and anti-inflammatory studies were obtained on a 680-Bio-Rad microplate reader (serial number 14966, Irvine, CA, USA).

### 2.2. Plant Material

*Eugenia uniflora* leaves were collected at the cultivated site in the Faculty of Agriculture, Obafemi Awolowo University (O.A.U.), Ile-Ife, Nigeria (GPS coordinates: N 7°31′3.7992″, E 4°31′34.8528″) on 15 March 2021. The plant was authenticated at the Ife Herbarium, O.A.U., Ile-Ife, with a voucher number, IFE 16589. The leaves were air-dried under protection from direct sunlight in a screen house, until a moisture content of 5% was achieved for the raw material. The dried leaves were milled into powder and kept in an airtight plastic bag until further use.

### 2.3. Extraction and Fractionation

The leaf powder (1.0 kg) was extracted at 70 °C with 5.0 L of 80% EtOH under reflux for 3 h. It was allowed to cool, filtered, and concentrated to dryness in vacuo on a Heidolph 110 Laborata rotavapor (Heidolph Instruments GmbH & Co. KG, Schwabach, Germany). The concentrated extract was kept inside a desiccator under silica for 72 h to remove water from the organic extract, thus affording the leaf EtOH extract. The extract (114.3 g) was suspended in 417 mL of distilled water and successively partitioned with organic solvents in increasing order of polarity to afford *n*-hexane (2 × 500 mL, 8.1% yield), dichloromethane (DCM, 3 × 500 mL, 22.1% yield), ethyl acetate (EtOAc, 4 × 500 mL, 17.9% yield) and aqueous (45.7% yield) fractions. The solvent-partitioning process was TLC-monitored to ensure an exhaustive transfer of the organic fractions from the aqueous phase. Finally, the fractions were concentrated in vacuo and kept inside a desiccator under silica to ensure complete dryness until further use.

### 2.4. Isolation of Compounds

The EtOAc fraction (17.5 g) was further fractionated because it demonstrated the best antioxidant activity among the partition fractions. It was adsorbed onto 35 g of silica gel and eluted on a 175 g silica gel column. The column was eluted with 500 mL of each of the following solvent systems, in increasing order of polarity, to afford one-hundred and fifty seven (157) collected fractions: *n*-hexane, *n*-hexane-EtOAc (90:10, 80:20, 70:30, 60:40, 50:50, 40:60, 30:70, 20:80 and 10:90), EtOAc, EtOAc-MeOH (95:5, 90:10, 80:20, 70:30, 50:50 and 30:70) and MeOH. The eluates were bulked into seven subfractions (F1–F7) based on their TLC patterns. The subfractions were screened for the best free radical scavenging fraction by spraying the developed TLC plates with 4 mg/mL DPPH solution in methanol. Subfraction F5 presented the strongest bleaching reaction, with yellow spots against a purple DPPH background after 5 min of incubation in the dark. Thus, it was further fractionated on a Sephadex LH-20 column by an isocratic elution process with DCM-MeOH (70:30). This afforded forty-one eluates, which were further categorized into five-column-bulked subfractions F5(i–v) after TLC analysis. Subfractions F5ii and F5iv exhibited yellowish ($R_f$ 0.49, 51 mg) and bright yellowish ($R_f$ 0.48, 29 mg) single spots, respectively, after they were developed in a TLC system of DCM-MeOH (70:30), followed by activation with 10% sulfuric acid spray. The yellow coloration appeared to char (darkened) after 2 min of heating at 105 °C, suggesting them to be glycosidic compounds. The two subfractions strongly bleached the purple DPPH reagent, suggesting them to have free radical scavenging properties. They exhibited 196–198 °C and 197–199 °C melting point ranges, respectively, when analysed on a Gallenkamp MPD350-BM 3.5 electrothermal instrument (Gallenkamp, Kent, UK). Hence, they were coded as compounds **1** and **2**, respectively.

### 2.5. Nuclear Magnetic Resonance (NMR) Analysis

The $^1$H-, $^{13}$C- and 2D NMR spectral data of the isolated compounds were acquired in DMSO-$d_6$ (Sigma-Aldrich Chemie GmbH, Taufkirchen, Germany) and recorded on an Avance 600 MHz Spectrometer (Bruker Biospin GmbH-Rheinstetten, Germany), which is situated at the Central Analytical Facilities, Stellenbosch University, Stellenbosch, South Africa. Tetramethyl silane (TMS) was used as an internal standard, while chemical shifts were recorded in part per million (ppm).

### 2.6. High-Resolution Mass Spectrometric (HRESI-MS) Analysis

The mass spectra of compounds were recorded on a Quadrupole Time-of-Flight (QToF) Synapt G2 Mass Spectrometer (Waters Corporation, Milford, MA, USA). The mass determination was achieved via electrospray ionization method (HRESI-MS). The mass-to-charge ratios were acquired in the positive ion mode and over a scan range of $m/z$ 100–1000.

### 2.7. Cytotoxicity Study

2.7.1. Cell Culture

The Hep-G2 and HeLa cells were supplied by Highveld Biologicals (Pty) Ltd., Lyndhurst, South Africa. The non-transformed HEK-293 cells (control cells) were supplied by the University of Witwatersrand, Medical School, South Africa. Eagle's Minimal Essential Medium (EMEM) with L-glutamine—Lonza BioWhittaker (Verviers, Belgium). Foetal bovine serum (FBS)—HyClone UK Ltd. (Cramlington, Northumberland, UK). Penicillin/streptomycin mixture (10,000 U/mL penicillin, 10,000 µg/mL streptomycin)—Lonza BioWhittaker (Verviers, Belgium). Sterile plasticware for cell culture—Corning Inc. (Corning, NY, USA). The 3-(4,5-dimethylthiazol-2-yl)-2,5-diphenyltetrazolium bromide (MTT) salt and dimethylsulphoxide (DMSO) were purchased from Merck, Darmstadt, Germany.

2.7.2. MTT Assay

The cytotoxicity of the extract, partition fractions and isolated compounds against HEK-293, Hep-G2 and HeLa cells was determined, using the MTT assay method, as previously described by Jagaran and Singh [22]. The three cell lines were propagated in growth medium (EMEM supplemented with 10% (*v/v*) FBS, 100 U/mL penicillin, and 100 µg/mL streptomycin). Cells were seeded at an average density of 25,000 cells/well in 96-well plates. This was maintained at 37 °C for 24 h to reach semi-confluency. The test samples were dissolved in 10% (*v/v*) DMSO with brief vortexing and sonication. Stock concentrations were as follows: 1, 2.5, 5, 7.5 and 10 µg/µL. Cells were prepared by draining the wells and adding fresh medium (100 µL/well). The test samples (10 µL) were introduced to result in final concentrations of 10, 25, 50, 75 and 100 µg/well. The final concentration of DMSO to which treated cells were exposed was 1% (*v/v*). Treatments were performed in triplicate. Untreated cells were included as positive (100% cell viability) controls. Cells treated with 10% (*v/v*) DMSO (10 µL/well) served as additional controls. Cells were incubated at 37 °C for 48 h. Growth medium was aspirated and cells were incubated (37 °C, 4 h) per well with 100 µL of the medium and MTT solution (5 mg/mL in phosphate buffered saline). Wells were drained, and formazan crystals were dissolved in DMSO (100 µL/well) to result in purple-coloured solutions. Absorbance was read at 540 nm in a Mindray microplate reader, MR 96A (Vacutec, Hamburg, Germany), against pure DMSO as a blank. The percentage cell viability was calculated as follows:

$$\frac{[A_{540nm}(\text{treated cells}) - A_{540nm}(\text{blank})]}{[A_{540nm}(\text{untreated cells}) - A_{540nm}(\text{blank})]} \times 100$$

The concentration that showed 50% cytotoxicity (reduced the viability of each cell line by 50%) was determined as the $CC_{50}$ value of each test sample.

### 2.8. Antioxidant Study

2.8.1. DPPH Radical Scavenging Assay

The DPPH assay was conducted by following a previously reported method [23]. A total volume of 0.5 mL of 0.1 mM DPPH free radical in methanol was added to 0.5 mL of serially diluted extract, partition fractions, isolated compounds and quercetin at 100.00, 50.00, 25.00, 12.50 and 6.25 µM concentration range in three replicates. The reaction mixture was incubated in the dark at 37 °C for 30 min. The absorbance was measured at 517 nm on a microplate reader. The percentage inhibition of the radical was calculated thus:

$$\% \text{ DPPH inhibition} = \left( \frac{\text{ABScontrol} - \text{ABSsample}}{\text{ABScontrol}} \right) \times 100$$

where ABSsample = absorbance of test sample, while ABScontrol = absorbance of negative control (methanol).

The final antioxidant activity was expressed as the $IC_{50}$ value, which represents the concentration that caused a 50% inhibition of the DPPH radical.

2.8.2. Nitric Oxide (NO) Radical Inhibition Assay

The ability of the extracts and isolated compounds to scavenge NO free radical was determined, using a previously described method [24]. The test samples were prepared in varying concentrations from 100.00 to 6.25 µg/mL and added to 0.2 mM (2 mL) of sodium nitroprusside in triplicates. The reaction mixture was incubated at 25 °C for 3 h. Then, 0.5 mL of the mixture was added to Griess reagent (0.33% sulphanilamide dissolved in 20% glacial acetic acid) and mixed with 1 mL of naphthylethylenediamine chloride (0.1% *w/v*). The afforded mixture was incubated at 25 °C for 30 min. Thereafter, the absorbance was measured at 540 nm on a microplate reader. The $IC_{50}$ of each test sample was also determined after the determination of their mean percentage inhibition, as thus:

$$\text{\% NO inhibition} = \left( \frac{\text{ABScontrol} - \text{ABSsample}}{\text{ABScontrol}} \right) \times 100$$

where ABScontrol = absorbance of control (methanol) and ABSsample = absorbance of extract/fractions/isolated compounds/quercetin/*L*-ascorbic acid.

2.8.3. Hydrogen Peroxide ($H_2O_2$) Inhibition Assay

The capacity of the extracts and isolated compounds to inhibit hydrogen peroxide free radical was determined, using a standard colorimetric method [25]. Here, the test samples (400 µL each) were serially diluted from 100 to 6.25 µM concentrations in triplicates. It was mixed with 60 µL of hydrogen peroxide solution (4 mM) prepared in a 0.1 M phosphate buffer saline (pH 7.4). The reaction mixture was incubated at 25 °C for 10 min, followed by a determination of the absorbance at 405 nm. The percentage inhibition of the hydrogen peroxide radical was calculated, using the formula:

$$\text{\% inhibition of hydrogen peroxide radical} = \left( \frac{\text{ABScontrol} - \text{ABSsample}}{\text{ABScontrol}} \right) \times 100$$

where ABScontrol = absorbance of control (methanol) and ABSsample = absorbance of extract/fractions/isolated compounds/quercetin/*L*-ascorbic acid.

*2.9. In Vitro Anti-Inflammatory Study*

The in vitro anti-inflammatory response and the optimum response time of the isolated compounds to act against protein denaturation was determined, using the egg albumin denaturation assay [26]. A reaction mixture comprising 0.2 mL of albumin content of fresh chicken egg, 2.8 mL of phosphate buffer saline at pH 6.4 and 2 mL each of the isolated compounds and diclofenac (standard drug) at 100, 50, 25, 12.5, 6.25 and 3.125 µg/mL concentrations was prepared in triplicate. The reaction mixture was incubated at 37 °C for 15 min away from direct light and thereafter boiled at 70 °C for 5 min in a thermostatic water bath. The resulting mixture was cooled, and the absorbance was measured at 655 nm. The experiment was repeated with compounds tested at a median concentration of 20 µg/mL and at a varied incubation period of 0, 1, 2, 3 and 4 h away from direct light and under room temperature (25 °C). The percentage inhibition of the compounds was calculated, using the formula:

$$\text{\% inhibition of EAD} = \left( \frac{\text{ABScontrol} - \text{ABSsample}}{\text{ABScontrol}} \right) \times 100$$

where EAD = egg albumin denaturation, ABSsample = absorbance of compounds/diclofenac and ABScontrol = absorbance of control (methanol).

*2.10. Statistical Analysis*

The cytotoxicity result was analysed on a Microsoft Excel version 365 (Microsoft® Corporation, Redmond, WA, USA). Data were expressed as the mean ± SD (n = 3). Sta-

tistical analysis was via One-Way Analysis of Variance (ANOVA), followed by Student Newman–Keul's post hoc test, while 95% confidence limit ($p < 0.05$) was considered as the level of significance of the acquired data.

## 3. Results and Discussion

### 3.1. Characterization of Isolated Compounds

#### 3.1.1. Myricitrin (**1**), $C_{21}H_{20}O_{12}$

Isolated as a yellow amorphous solid, 51 mg, m.p. 196–198 °C.

HRESI + TOF–MS (*m/z*, % rel. int.): *m/z* 465.1022 $[M + H]^+$ (100%), calculated for $C_{21}H_{20}O_{12}$: 465.1033; 319.0441 $[M - C_6H_{11}O_5 + H]^+$ (89%), calculated for $C_{15}H_{11}O_8$: 319.0454; $^1H$ NMR (600 MHz, DMSO-$d_6$) $\delta_H$: 0.85 (3H, *d*, *J* = 6.1 Hz, H-6″), 3.15 (1H, *t*, *J* = 9.3 Hz, H-4″), 3.36 (1H, *dq*, *J* = 5.1, 6.5 Hz, H-5″), 3.56 (1H, *dd*, *J* = 2.5, 4.5 Hz, H-3″), 3.98 (1H, *d*, *J* = 3.3 Hz, H-2″), 5.20 (1H, *s*, H-1″), 6.20 (1H, *d*, *J* = 2.5 Hz, H-6), 6.37 (1H, *d*, *J* = 2.5 Hz, H-8), 6.91 (2H, *d*, *J* = 6.9 Hz, H-2′, H-6′); $^{13}C$ NMR (150 MHz, DMSO-$d_6$) as presented in Table 1. The spectra of myricitrin (**1**) are available in Supplementary Materials (Figures S1 and S2).

#### 3.1.2. Unifloratrin (**2**), $C_{21}H_{20}O_{12}$

Isolated as a golden-yellow amorphous solid, 29 mg, m.p. 197–199 °C; HRESI + TOF– MS (*m/z*, % rel. int.): *m/z* 487.0907 (1%) $[M + Na]^+$, calculated for $C_{20}H_{20}O_{12}.Na$: 487.0708; 465.1042 $[M + H]^+$ (100%), calculated for $C_{21}H_{20}O_{12}$: 465.1033, 319.0451 $[M - C_6H_{11}O_5 + H]^{+.}$ (81%), calculated for $C_{15}H_{11}O_{12}$: 319.0454; $^1H$ NMR (600 MHz, DMSO-$d_6$) $\delta_H$: 0.85 (3H, *d*, *J* = 6.3 Hz, H-6″), 3.15 (1H, *t*, *J* = 9.3 Hz), 3.36 (1H, *dq*, *J* = 5.2, 6.9 Hz, H-5″), 3.56 (1H, *dd*, *J* = 2.5, 4.4 Hz, H-3″), 3.98 (1H, *d*, *J* = 3.3 Hz, H-2″), 5.20 (1H, *s*, H-1″), 6.21 (1H, *d*, *J* = 2.2 Hz, H-6), 6.37 (1H, *d*, *J* = 2.6 Hz, H-8), 6.88 (1H, *d*, *J* = 2.0 Hz, H-6′), 6.92 (1H, *d*, *J* = 1.9 Hz, H-3′), 12.68 (H, *s*); $^{13}C$ NMR (150 MHz, DMSO-$d_6$) as presented in Table 1. The spectra of unifloratrin (**2**) are available in Supplementary Materials (Figures S3 and S4).

### 3.2. Structure Elucidation of Isolated Compounds

The NMR spectra of myricitrin (**1**) and unifloratrin (**2**) showed similar patterns. Both compounds exhibited one methyl proton and five oxygenated methine protons between 3.15 and 5.20 ppm, suggesting a flavonoid with a rhamnose sugar unit, four olefinic protons at 6.21 ppm (H-5) and 6.37 ppm (H-7), and two non-equivalent olefines at 6.88 ppm and 6.91 ppm. The narrow singlet signal (hump) at 12.68 ppm suggests an intermolecular hydrogen bond between the proton of the hydroxyl signal (5-OH) with the oxygen of the neighbouring carbonyl at position C-4, thus confirming both compounds to be flavonol glycosides [27]. The molecular structures of the compounds are presented in Figure 1, while the difference in the compounds was revealed via the COSY and HMBC experiments, as illustrated in Figure 2. The COSY spectrum of unifloratrin (**2**) showed direct proton correlations ($^1H$–$^1H$) of the rhamnose unit, that is, 3.56 ppm (H-3″) correlated with 3.98 ppm (H-2″) and 3.16 ppm (H-4″), while 3.36 ppm (H-5″) correlated with 3.16 ppm (H-4″) and 0.85 ppm (H-6″, methyl), in comparison with literature report [27]. The HMBC spectrum of unifloratrin (**2**) showed a three-bond correlation ($^3J_{CH}$) between the proton signal (H-3′) at 6.92 ppm and the carbon signal (C-5′) at 167.5 ppm (Figure 2). Also, a three-bond correlation was observed between the proton signal (H-6′) at 6.88 ppm and the carbon signal (C-2′) at 136.5 ppm, suggesting a 2,4,5-trihydroxyphenyl ring B of the flavonoid nucleus (Figure 1). The $^{13}C$ NMR signals at 145.8 ppm and 167.5 ppm were ascribed to C-4′ and C-5′ positions. These downfield signals might be due to the conjugation of the B ring with the carbonyl group at position C-4, leading to an increased pi-electron density at positions C-2′, C-4′ and C-6′ [28].

**Figure 1.** Molecular structures of myricitrin (**1**) and unifloratrin (**2**) isolated from *Eugenia uniflora* leaves.

Unifloratrin (**2**)

**Figure 2.** COSY ($^1$H–$^1$H) and key HMBC (C→H) correlations of unifloratrin (**2**).

The molecular structure of unifloratrin (**2**) was confirmed from their HR-MS data (Figure 3), which showed quasi-molecular ion peaks [M + H]$^+$ at *m/z* 465.1042, having the same calculated mass with myricitrin (**1**) at *m/z* 465.1033 ($C_{21}H_{20}O_{12}$). In addition, unifloratin (**2**) showed a sodiated molecular ion peak at *m/z* 487.0907 [M + Na]$^+$, which is consistent with the molecular formula, $C_{21}H_{20}O_{12}$. The presence of a rhamnopyranoside unit was confirmed by the fragment ions at *m/z* 319.0451, which might be due to the homolytic cleavage of rhamnose [M − 146 + H]$^+$, to afford myricetin [29].

Thus, based on the observed mass and 1D and 2D NMR spectral data (Table 1 and Figure 1) and in comparison with the spectral data of myricitrin reported from the leaves of *Newtonia buchananii* [30], *Albizia amara* [31], *Elaeocarpus floribundus* [32] and the flowers of the African water lily, *Nymphaea caerulea* [33], the compounds were elucidated as myricetin-3-O-α-L-rhamnopyranoside, also known as myricitrin (**1**), and its newly described isomer, 5,7-dihydroxy-3-(3,4,5-trihydroxy-6-methyltetrahydropyran-2-yloxy)-2-(2,4,5-trihydroxyphenyl)chromen-4-one, named "unifloratrin (**2**)".

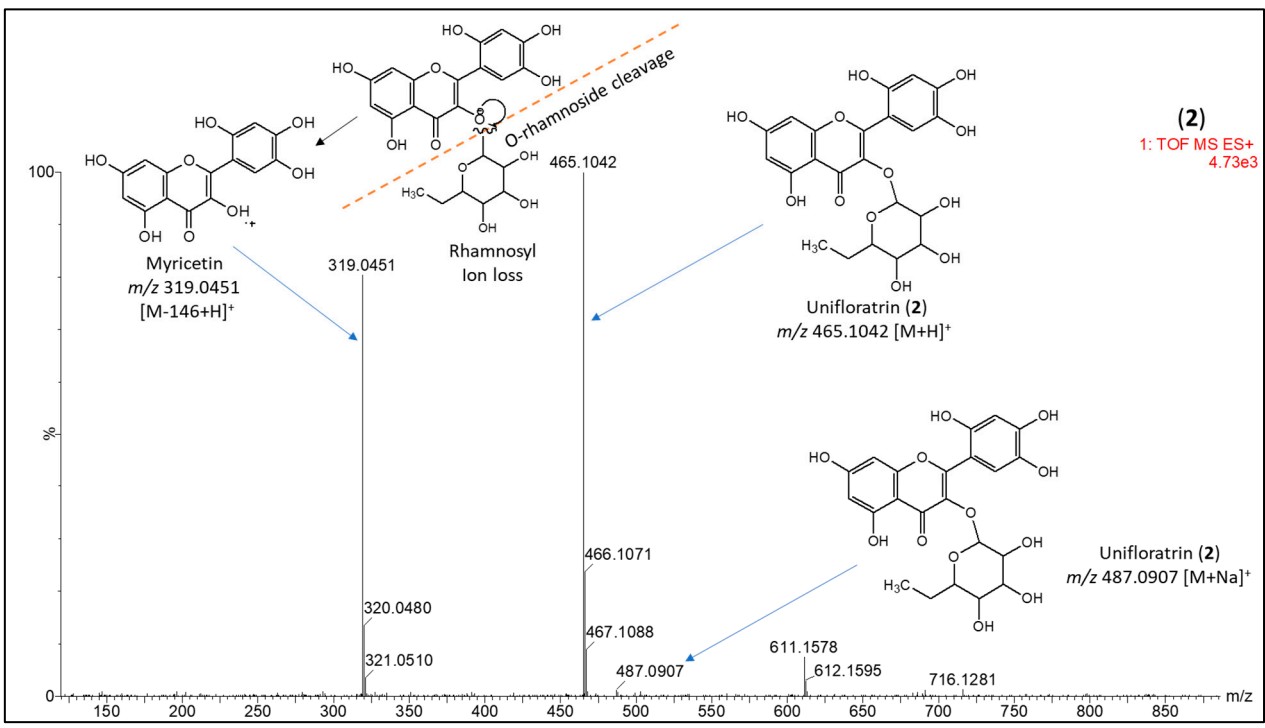

**Figure 3.** HRESI-MS fragmentation pattern of unifloratrin (**2**) isolated from *E. uniflora* leaves.

**Table 1.** NMR spectral data of myricitrin (**1**) and unifloratrin (**2**) isolated from *Eugenia uniflora* leaves.

| Position | Unifloratrin (2) | | | | Myricitrin (1) | |
|---|---|---|---|---|---|---|
| | DEPT135 | $\delta_H$ | HMBC | $\delta_c$ | Acquired $\delta_c$ | Literature $\delta_c$ [31] |
| 2 | C | | | 157.6 | 157.5 | 157.0 |
| 3 | C | | | 134.3 | 134.3 | 134.7 |
| 4 | C=O | | | 177.8 | 177.8 | 178.1 |
| 4a | C | | | 104.1 | 104.1 | 104.2 |
| 5 | C | | | 161.4 | 161.4 | 161.7 |
| 6 | CH | 6.21, 1H, d, *J* = 2.2 Hz | C-4a, C-8 | 98.5 | 98.7 | 99.1 |
| 7 | C | | | 164.2 | 164.2 | 165.6 |
| 8 | CH | 6.37, 1H, d, *J* = 2.6 Hz | C-4a, C-6 | 93.3 | 93.6 | 94.6 |
| 8a | C | | | 156.5 | 156.5 | 157.8 |
| - | | 12.68, s | | | | |
| 1′ | C | | | 120.5 | 119.6 | 120.1 |
| 2′ | C | | | 136.5 | 107.9 | 108.4 |
| 3′ | CH | 6.92, 1H, d, *J* = 1.9 Hz | C-5′ | 108.8 | 145.8 | 146.3 |
| 4′ | C | | | 145.8 | 136.5 | 137.0 |
| 5′ | C | | | 167.5 | 145.8 | 146.3 |
| 6′ | CH | 6.88, 1H, d, *J* = 2.0 Hz | C-2, C-2′, C-4′ | 107.9 | 107.9 | 108.4 |
| | L-Rhamnose | | | | | |
| 1″ | CH | 5.20, 1H, s | C-3″ | 102.0 | 101.9 | 102.4 |
| 2″ | CH | 3.98, 1H, d, *J* = 3.3 Hz | | 70.1 | 70.0 | 71.8 |
| 3″ | CH | 3.56, 1H, dd, *J* = 2.5, 4.4 Hz | | 70.4 | 70.4 | 70.9 |
| 4″ | CH | 3.15, 1H, t, *J* = 9.3 Hz | C-2″, C-6″ | 71.3 | 71.3 | 71.0 |
| 5″ | CH | 3.36, 1H, dq, *J* = 5.2, 6.9 Hz | | 70.6 | 70.6 | 70.4 |
| 6″ | CH₃ | 0.85, 3H, d, *J* = 6.3 Hz | C-4″ | 17.6 | 17.5 | 18.0 |

Methyl ($CH_3$); methine (CH); quaternary carbon (C); carbonyl (C=O); singlet (s); doublet (d); triplet (t); doublet of a quartet (dq); coupling constant (J) is expressed in Hertz (Hz); chemical shifts ($\delta_H$ and $\delta_C$) are expressed in part per million (ppm).

*3.3. Evaluation of Cytotoxicity*

It was observed in the MTT test that the 10% (*v*/*v*) DMSO used as an additional control demonstrated ≥99.9% cell viability, which suggests that the solvent (vehicle) did not have any significant impact on the cytotoxicity of the solutes (extracts, fractions and isolated compounds). The cytotoxicity result (Table 2) showed that the EtOAc fraction of *E. uniflora* exhibited the best activity among the partition fractions, with $CC_{50}$ values of $28.2 \pm 3.1$ and $25.6 \pm 3.4$ μg/100 μL against Hep-G2 and HeLa cancer cells, respectively, and the least active against HEK-293 (normal) cells with a $CC_{50}$ of $32.5 \pm 6.1$ μg/100 μL. Recent findings have shown the *n*-hexane and EtOAc fractions of *E. uniflora* leaves with considerable potency against human breast cancer JIMT-1 cells at $CC_{50}$ values of 151.9 and 285.6 μg/mL, respectively [10]. Another study conducted on a closely related species, *E. polyantha*, showed that the leaf EtOAc fraction is flavonoid-rich (5.3 mgQE/g), with a $CC_{50}$ of 171.9 μg/mL against human breast cancer T47D cells [34]. Therefore, these findings justified our focus on the EtOAc fraction of *E. uniflora* for its bioactive compounds.

**Table 2.** Cytotoxic activity of extract, partition fractions and compounds isolated from *Eugenia uniflora* leaves.

| Test Sample | $CC_{50} \pm$ SD (μg/100 μL) | | |
| --- | --- | --- | --- |
| | **HEK-293** | **Hep-G2** | **HeLa** |
| EtOH | $38.2 \pm 3.3$ [bc] | $30.8 \pm 2.9$ [cd] | $32.6 \pm 4.1$ [d] |
| *n*-Hexane | $42.5 \pm 6.8$ [c] | $35.4 \pm 4.3$ [d] | $33.8 \pm 3.0$ [d] |
| DCM | $42.3 \pm 3.9$ [c] | $35.2 \pm 3.0$ [d] | $31.3 \pm 2.8$ [d] |
| EtOAc | $32.5 \pm 6.1$ [bc] | $28.2 \pm 3.1$ [cd] | $25.6 \pm 3.4$ [cd] |
| Aqueous | $38.0 \pm 4.7$ [c] | $29.7 \pm 4.0$ [cd] | $30.9 \pm 5.7$ [cd] |
| Myricitrin (**1**) | $27.3 \pm 3.1$ [b] | $22.0 \pm 2.7$ [ab] | $8.5 \pm 2.2$ [a] |
| Unifloratrin (**2**) | $27.6 \pm 3.2$ [b] | $25.5 \pm 4.9$ [bc] | $14.8 \pm 2.0$ [b] |
| 5-FU | $6.1 \pm 1.2$ [a] | $17.5 \pm 2.1$ [a] | $7.5 \pm 1.5$ [a] |

Data are presented as mean $\pm$ S.D. (*n* = 3); different letters mean significant differences (*p* < 0.05).

An evaluation of the potency of myricitrin and unifloratrin against the non-tumourigenic HEK-293 cells (Table 2) showed that they were comparable (*p* > 0.05) in activity, with a $CC_{50}$ of 27 μg/100 μL, while 5-FU exhibited a significantly lower activity (*p* < 0.05), with a $CC_{50}$ of $6.1 \pm 1.2$ μg/100 μL. Thus, each of myricitrin (**1**) and unifloratrin (**2**) can be regarded as less toxic to normal cells compared to 5-FU in this study.

An assessment of the compounds against cancerous cells showed that they possess comparable cytotoxicity (*p* > 0.05) against Hep-G2 cells, with a $CC_{50} \approx 22.0$ μg/100 μL. However, this activity was lower compared to 5-FU, which exhibited a $CC_{50}$ of $17.5 \pm 2.1$ μg/100 μL. Furthermore, myricitrin (**1**) showed better potency against the HeLa cells when compared to unifloratrin (**2**), based on its significantly lower (*p* < 0.05) $CC_{50}$ of $8.5 \pm 2.2$ μg/100 μL. It is worthy of mention that, the observed activity of myricitrin (**1**) was comparable (*p* > 0.05) to that of 5-FU ($CC_{50} = 7.5 \pm 1.5$ μg/100 μL).

The level of potency of isolated compounds was also expressed in this study as mean percentage cell survival at various test concentrations. The results presented in Figure 4 showed that there was concentration-dependent decrease in the mean percentage survival of Hep-G2 and HeLa cells at 10–100 μg/100 μL concentrations. Myricitrin (**1**), unifloratrin (**2**) and 5-FU all showed comparable (*p* > 0.05) reduction in the Hep-G2 cells at 50, 75 and 100 μg/100 μL, rspectively. There was also a comparable (*p* > 0.05) reduction in the HeLa cells when treated with myricitrin (**1**) and 5-FU at 10, 25, 50 and 100 μg/100 μL concentrations.

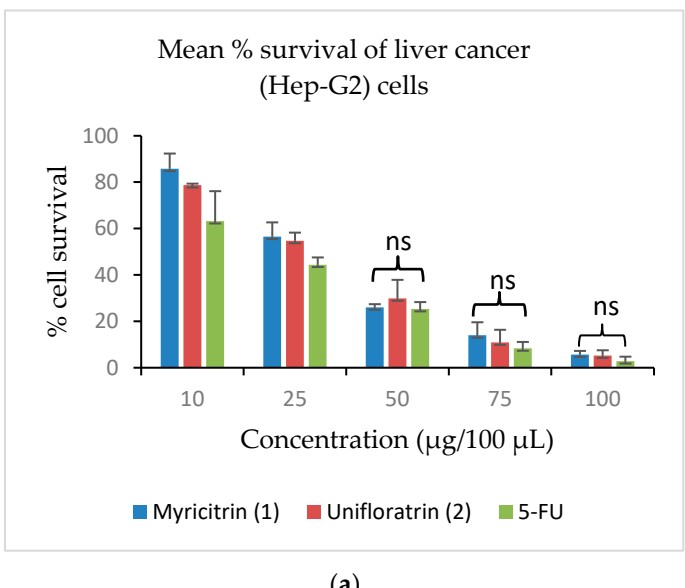

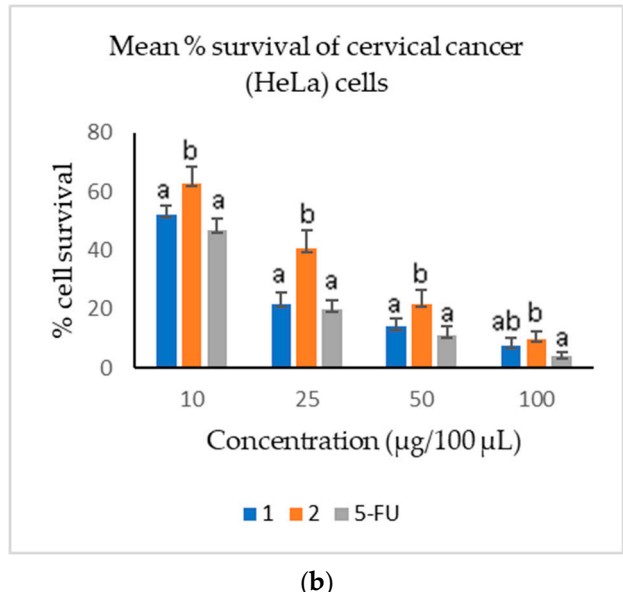

<div align="center">(<b>a</b>)</div>

<div align="right">(<b>b</b>)</div>

**Figure 4.** Cytotoxic activity of myricitrin (**1**) and unifloratrin (**2**) isolated from *Eugenia uniflora* leaves. Symbol ns in (**a**) indicates no significant difference ($p > 0.05$); letters a, b and ab in (**b**) indicate significant differences ($p < 0.05$).

Myricitrin (2.5–10 µg/mL) has been reported to show significant inhibitory effects of on acrylamide-mediated oxidative stress and cytotoxicity in human gastrointestinal Caco-2 cells [35]. Also, myricitrin isolated from the leaves of *Madhuca longifolia* has been reported to inhibit the proliferation of acute myelogenous leukaemia cells HL-60 in an apoptotic-dependent manner, with an $IC_{50}$ of 353 µM [36].

Evidence has shown that the number of hydroxyl groups and their positions play a considerable role in the level of stability and cytotoxicity of flavonoids [37]. In specific terms, the presence of hydroxyl groups at positions C-3′ and C-4′ or C-4′ and C-5′ of flavonoid ring B has been reported to enhance the distribution of electron cloud around the phenyl ring of flavonoids, which in turn makes them more liable to donate protons to form hydrogen bonds with the cell active site (conjugation), to inhibit oxidative and/or biological activities [38]. Therefore, the higher activity of myricitrin (**1**) as compared to unifloratrin (**2**) in this study may be justified by the positional difference in the trihydroxyphenyl substituents of the former at positions C-3′, C-4′ and C-5′ on the ring B of the flavonoid.

### 3.4. Evaluation of Antioxidant Activity

According to the result in Table 3, the ethyl acetate fraction of *E. uniflora* was more active than the mother extract. Also, each of the characterized compounds exhibited more antioxidant activity than the mother EtOAc fraction.

Myricitrin (**1**) exhibited a double-fold $H_2O_2$ radical scavenging activity, with an $IC_{50}$ of $30.46 \pm 1.79$ µM when compared to quercetin ($IC_{50} = 61.17 \pm 2.58$ µM). On the other hand, *L*-ascorbic acid demonstrated the best NO inhibitory activity among the test samples, with an $IC_{50}$ of $21.21 \pm 1.44$ µM. Further evaluation showed that myricitrin (**1**) exhibits DPPH ($6.23 \pm 1.09$ µM) and NO ($22.01 \pm 2.59$ µM) inhibitory activities, which were comparable ($p > 0.05$) to quercetin at $IC_{50}$ values of $7.11 \pm 1.55$ and $25.54 \pm 1.27$ µM, respectively. It also demonstrated a significantly better ($p < 0.05$) inhibitory activity when compared to unifloratrin (**2**). Based on these findings, both compounds could only have exhibited these radical scavenging activities because of their polyphenolic structures, which enable them to easily transfer hydrogen atoms through a mechanism of hydrogen atom transfer, while also making electrons readily available for easy abstraction by free radicals via a mechanism of single electron transfer [39].



**Table 3.** In vitro antioxidant activity of *Eugenia uniflora* leaf extracts and isolated compounds.

| Test Sample | IC$_{50}$ ± SD (µM) | | |
| --- | --- | --- | --- |
| | **DPPH** | **NO** | **H$_2$O$_2$** |
| EtOH | 33.28 ± 3.17 [d] | 57.23 ± 3.16 [e] | 123.37 ± 11.05 [f] |
| Hexane | 79.14 ± 8.57 [e] | 101.87 ± 12.30 [f] | 379.19 ± 22.81 [g] |
| DCM | 38.91 ± 4.45 [d] | 51.02 ± 4.22 [e] | 112.74 ± 5.29 [ef] |
| EtOAc | 23.52 ± 4.61 [c] | 39.16 ± 3.56 [d] | 87.50 ± 4.44 [d] |
| Aqueous | 31.08 ± 3.87 [d] | 45.25 ± 2.92 [de] | 104.27 ± 4.01 [e] |
| Myricitrin (**1**) | 6.23 ± 1.09 [a] | 22.01 ± 2.59 [a] | 30.46 ± 1.79 [a] |
| Unifloratrin (**2**) | 12.53 ± 3.36 [b] | 34.10 ± 3.69 [c] | 57.42 ± 3.61 [b] |
| *L*-ascorbic acid | 8.32 ± 1.69 [a] | 21.21 ± 1.44 [a] | 54.92 ± 2.45 [b] |
| Quercetin | 7.11 ± 1.55 [a] | 25.54 ± 1.27 [ab] | 61.17 ± 2.58 [c] |

Data are expressed as mean ± SD (*n* = 3); extract (EtOH), partition fractions (hexane, DCM, EtOAc, aqueous); different letters indicate significant differences (*p* < 0.05).

Myricitrin has been reported to possess considerable DPPH, NO and H$_2$O$_2$ radical scavenging activity, with IC$_{50}$ values of 1.31, 21.54 and 28.46 µg/mL [40]. It has also been shown to exhibit in vivo antioxidant activity as a NO and protein kinase C inhibitor, with stronger free radical scavenging activity than quercetin and other flavonol rhamnosides [41]. A justification for the better antioxidant activity observed in myricitrin (**1**) when compared to unifloratrin (**2**) could be the effective radical scavenging properties that have been reported for flavonoids with trihydroxy functions, especially at positions 3′, 4′ and 5′ [38,42].

*3.5. Evaluation of In Vitro Anti-Inflammatory Activity*

The ability of the isolated compounds to inhibit protein denaturation was captured in this study as a measure of anti-inflammatory response of the compounds, in vitro. The result presented in Figure 5 shows the level of inhibition of the compounds and their optimal response time. The result shows a concentration-dependent decrease in the level of inhibition of myricitrin (**1**), unifloratrin (**2**) and the standard anti-inflammatory drug (diclofenac) from 100 to 3.125 µg/mL. At the highest concentration, myricitrin and diclofenac demonstrated a comparable activity (*p* > 0.05) of about 95% inhibition of EAD, while at 6.25 and 3.125 µg/mL, the activity of both compounds was comparable (*p* > 0.05).

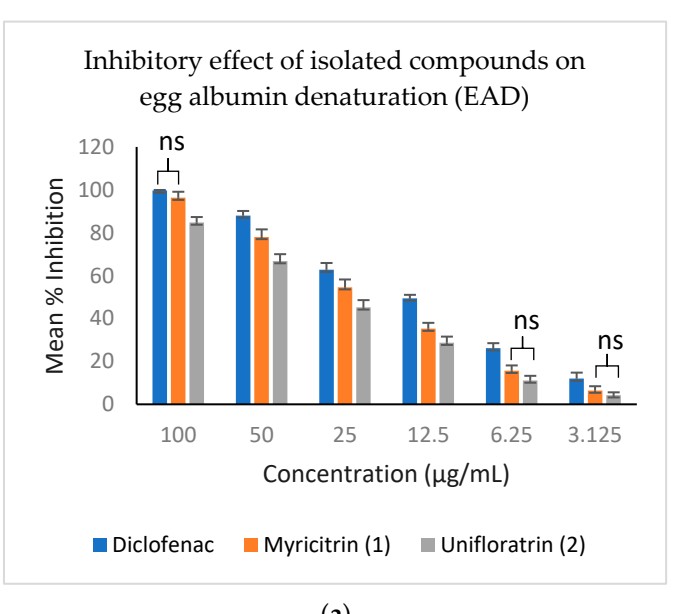

(**a**)

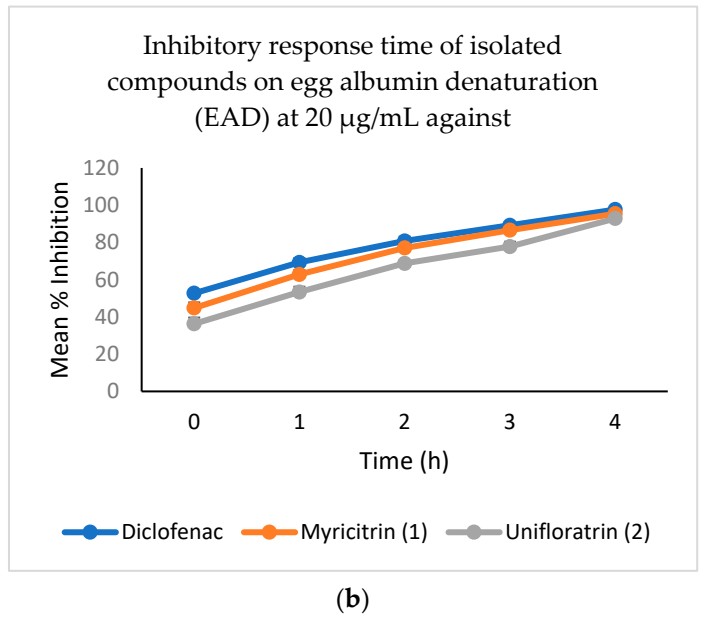

(**b**)

**Figure 5.** In vitro anti-inflammatory activity of myricitrin (**1**) and unifloratrin (**2**) isolated from *Eugenia uniflora* leaves. Symbol ns represents no significant difference (*p* > 0.05).

Furthermore, it was observed that the inhibitory effects of myricitrin (**1**) and diclofenac were comparable at 20 μg/mL and 2 h incubation time. Meanwhile, at 20 μg/mL and under 4 h of incubation, unifloratrin (**2**) showed a comparable ($p > 0.05$) inhibitory effect on egg albumin denaturation (EAD) to both myricitrin (**1**) and diclofenac. These findings have further shown the significance of dose–response time to the assessment of the anti-inflammatory properties of natural products.

Generally, flavonoids have been reported to have considerable anti-inflammatory properties by inhibiting cytokine release from RAW264.7 cells [43]. Myricitrin has been shown to exhibit in vivo anti-inflammatory activity by reducing the overexpression of inducible nitric-oxide synthase (iNOS) and nuclear factor-κB activation induced by lipopolysaccharide on RAW264.7 cells [44]. Myricitrin was shown to significantly reduce the overexpression of cyclooxygenase-2 (COX-2) and tumour necrosis factor-alpha (TNF-α) in the liver, suggesting the suppression of inflammation [40]. In another study, it attenuated LPS-mediated neuroinflammation by blocking the activation of the mitogen-activated protein kinase (MAPK) and nuclear factor kappa-B (NF-κB) signalling pathways in mice [45]. Myricitrin has also been shown to cause a significant reduction in tumour necrosis factor–α (TNF-α), stimulating the production of vascular cell adhesion protein-1 (VCAM-1) and intercellular adhesion molecule-1 (ICAM-1) by inhibition of the NF-κB pathways [46]. A closely related flavonol glycoside, myricetin-3-O-β-D-galactopyranoside, was reported to inhibit ultraviolet (UVA)-induced inflammation in vitro by suppressing pro-inflammatory cytokines through the inhibition of MAPK signalling and activation of TGFβ signalling pathways [47].

## 4. Conclusions

Two flavonol glycosides, myricitrin (**1**) and its newly described isomer, named unifloratrin (**2**), were isolated and characterized from the ethyl acetate fraction of *E. uniflora* leaves. The compounds showed considerable cytotoxicity against human cervical (HeLa) and liver (Hep-G2) cancer cells, in vitro, with myricitrin (**1**) demonstrating better activity than unifloratrin (**2**) in HeLa cells. The considerable cytotoxicity of the compounds could be due to their observed DPPH, NO and $H_2O_2$ radical scavenging activities, as well their ability to inhibit inflammatory action, such as protein denaturation. Thus, the study findings may serve as a justification for the extensive ethnomedicinal use of the plant, while flavonol glycosides such as myricitrin (**1**) may be further exploited as a candidate cytotoxic agent, especially against HeLa cells.

**Supplementary Materials:** The following supporting information can be downloaded at: https://www.mdpi.com/article/10.3390/scipharm91030042/s1, Figure S1: Mass spectrum of myricitrin (**1**); Figure S2: NMR spectra of myricitrin (**1**); Figure S3: mass spectrum of unifloratrin (**2**); Figure S4: NMR spectra of unifloratrin (**2**).

**Author Contributions:** Conceptualization, A.O.O. (Ayodeji Oluwabunmi Oriola); methodology, investigation, A.O.O. (Ayodeji Oluwabunmi Oriola), G.M.M. and M.S.; software, data curation, validation and formal analysis, A.O.O. (Ayodeji Oluwabunmi Oriola) and M.S.; resources, visualization and supervision, A.O.O. (Adebola Omowunmi Oyedeji); writing—original draft preparation, A.O.O. (Ayodeji Oluwabunmi Oriola); writing—review and editing, A.O.O. (Adebola Omowunmi Oyedeji); project administration and funding acquisition, A.O.O. (Adebola Omowunmi Oyedeji) and M.S. All authors have read and agreed to the published version of the manuscript.

**Funding:** The APC was funded by Directorate of Research Development and Innovation, Walter Sisulu University, South Africa.

**Institutional Review Board Statement:** Not applicable.

**Informed Consent Statement:** Not applicable.

**Data Availability Statement:** Not applicable.

**Acknowledgments:** The authors acknowledge the Walter Sisulu University for postdoctoral research fellowship award to the first author. J.A. Aladesanmi is appreciated for his pioneer work on *Eugenia uniflora*.

**Conflicts of Interest:** The authors declare no conflict of interest.

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
