# Peer review of "Flavonol Glycosides from Eugenia uniflora Leaves and Their In Vitro Cytotoxicity, Antioxidant and Anti-Inflammatory Activities"

_scipharm, doi:10.3390/scipharm91030042_

Round 1
Reviewer 1 Report
Dear Authors,
I present a review of the manuscript «Flavonol Glycosides from Eugenia uniflora Leaves and Their In Vitro Cytotoxicity, Antioxidant and Anti-inflammatory Activities».
Short summary:
This work corresponds to the subject of the journal. The scientific novelty lies in the phytochemical study of Eugenia uniflora leaves, and the study of the anti-inflammatory, antioxidant, cytotoxic activity of ethanol extract, individual fractions and individual compounds isolated from it. Two flavonolramnosides were isolated using column chromatography, and their chemical structure was determined using physico-chemical analysis methods. It was revealed that these substances have cytotoxicity comparable to 5-fluorouracil and their anti-inflammatory effect is comparable to diclofenac. The relationship between the antioxidant activity of flavonolramnosides and their chemical structure has been established. The experimental data obtained can serve as a justification for the ethnomedical use of the plant.
Comments on the general concept:
No comments
Specific comments
2.2. Plant Material
It is not entirely clear which leaves of the plant were used in the work, natural or cultivated.
Was the raw material protected from sunlight during drying?
What was the percentage of moisture content of the raw material after drying?
2.3. Extraction and Fractionation
Line 97: It is necessary to specify the concentration of ethanol.
Line 100: You need to specify the output of the extract.
How was the completeness of extraction by organic solvents monitored? You need to specify this information.
Why after extraction with ethyl acetate, butanol extraction was not carried out?
How was the removal of water from organic fractions carried out?
How were extracts, fractions and individual substances dried and stored?
2.4. Isolation of Compounds
Line 111: It is necessary to specify the ratio of fractions and sorbent Sephadex LH-20.
How was the individuality of the selected compounds checked?
2.8. Antioxidant Study
Why was quercetin chosen as a reference substance in the study of antioxidant activity?
3.2. Structure Elucidation of Isolated Compounds
Why has the presence of rhamnose in these compounds not been proven, for example, by hydrolysis and subsequent identification of the carbohydrate component?
How was the size of the rhamnose oxide ring determined?

Author Response
RESPONSES TO REVIEWER 1
2.2. Plant Material
It is not entirely clear which leaves of the plant were used in the work, natural or cultivated.
Was the raw material protected from sunlight during drying?
What was the percentage of moisture content of the raw material after drying?
Response:
The leaves used were obtained from a cultivated site, and this has now been stated in the revised manuscript.
The leaves were protected from direct sunlight by drying inside the screen house. This has been clearly stated.
The leaves were dried until moisture content of about 5% was achieved.
2.3. Extraction and Fractionation
Line 97: It is necessary to specify the concentration of ethanol.
Line 100: You need to specify the output of the extract.
How was the completeness of extraction by organic solvents monitored? You need to specify this information.
Why after extraction with ethyl acetate, butanol extraction was not carried out?
How was the removal of water from organic fractions carried out?
How were extracts, fractions and individual substances dried and stored?
Response:
It is necessary because it is not absolute or 99.9% EtOH. However, the statement in question has been modified to read “EtOH (80%).”
The output has been specified by mentioning that the extraction process gave an extract of 11.8 % yield.
The completeness of extraction was done by TLC analysis. This has now been stated in the revised manuscript.
n-BuOH was not used because it was not available. The supposed n-BuOH constituents can be harnessed by exploiting the afforded aqueous fraction.
The extract and fractions were concentrated in vacuo and then dried inside an activated desiccator until further use. This has been stated in the revised manuscript.
2.4. Isolation of Compounds
Line 111: It is necessary to specify the ratio of fractions and sorbent Sephadex LH-20.
How was the individuality of the selected compounds checked?
Response:
It may be necessary to state the ratios of solvents used (solvent systems) and adsorbents for reproducibility of results. The other reviewers subscribed to it.
The individuality of the compounds was checked based on TLC analysis as indicated in the revised paper.
2.8. Antioxidant Study
Why was quercetin chosen as a reference substance in the study of antioxidant activity?
Response:
Although, L-ascorbic acid was initially used as the antioxidant standard. After the isolated compounds were identified to be of the flavonoid class, we decided to test them against a flavonoid standard, hence, the choice of quercetin.
However, we have now added the antioxidant result for the L-ascorbic acid that was initially screened in the revised manuscript.
3.2. Structure Elucidation of Isolated Compounds
Why has the presence of rhamnose in these compounds not been proven, for example, by hydrolysis and subsequent identification of the carbohydrate component? How was the size of the rhamnose oxide ring determined?
We could not do further analysis such as acid hydrolysis and rhamnose oxide ring, to further justify the sugar unit.
However, we relied on high-resolution mass spectrometry (HRESI-MS) data to confirm cleavage of a rhamnosyl unit to give myricetin fragment ion (aglycone) at m/z 319. We have now illustrated this fragmentation data in the revised manuscript as Figure 3. This has been done in addition to the NMR spectral data and literature report.
Thank you.
Reviewer 2 Report
The subject of the present paper is interesting, the work is well designed, the obtained results are relevant, and seems to have novelty. The manuscripts is worth to be published provided that the issues pointed out below can be conveniently addressed.
General
The manuscript presents several minor typos that require correction. Some examples:
Line 33: Eugenia uniflora L must have a full stop after L.
Line 51: trans and cis must be italicized.
Line 102: n-hexane, the n must be italicized.
Line 169: 0.1 Mm.
Line 322, H2O2. The 2 must be subscript.
Line 324: “showed that 1”, 1 is not in bold.
Line334: in vivo must be italicized.
Line 346: in vitro must be italicized.
Once HEK-293, Hep-G2 and HeLa cell lines have been defined in the abstract, there is no need to repeatedly define the acronym of the cell lines throughout the manuscript.
The same for other acronyms such as MTT.
Abstract
Line 13: According to the abstract the authors performed an “activity-guided fractionation of the plant”. This is not truly the case. In fact, the authors performed the fractionation of the EtOH-distilled water 97 (4:1) extract of the plant leaves. Please correct.
Introduction
Line 67: The authors stated that flavonoids “are polyphenols synthesized by plants in response to microbial infections [20].” In fact flavonoids act as plant defence mechanisms against biotic and abiotic factors. Why the authors were reductive?
Results and discussion
General: m/z should be in italic. Conventionally, in deuterated solvents names the notation “d” is italicized as in DMSO-d6. In this case, “6” must be subscript.
Section 2.4.
Line 107: A “mixture” of n-hexane-EtOAc (100:0) is not a mixture, but a single solvent. A proportion is different from a percentage. Accordingly, hange “n-hexane-EtOAc (100:0, 90:10, 80:20, 70:30, 60:40, 50:50, 40:60, 30:70, 20:80 and 10:90%)” to “n-hexane, n-hexane-EtOAc (90:10, 80:20, 70:30, 60:40, 50:50, 40:60, 30:70, 20:80 and 10:90)”.
The same for the EtOAc-MeOH and DCM-MeOHmixtures.
Section 2.7.2.
Line 165: Define CC50.
Line 185: Define IC50.
Section 3
The authors say they performed an activity-guided fractionation of the EtOH-water extract of the plant leaves. However, there is no information on which activity served as the criteria to choose the EtOAc fraction to be further separated. Was it the cytotoxic activity, the antioxidant activity, both? The authors should clarify this.
In the same way, which was the criteria to choose subfraction F5 to be further fractionated by column chromatography? It is important to have the rational described.
Section 3.1.
Although both compounds 1 and 2 have 20 hydrogen atoms, the 1H NMR spectrum of compound 1 only records 11 hydrogen atoms, while that of compound 2 only records 12 hydrogen atoms. All hydrogen atoms must be attributed to the corresponding signals in the spectrum.
Line 225: Once compound 1 was identified and the molecular formula is given below “3.1.1. Compound 1 (myricitrin, C21H20O12)” should be replaced by “Myricitrin”.
Line 226: The yield provided is not a true yield as the plant’s content of myricitrin is not known. Therefore, it is misleading and should be removed. The same for compound 2.
Lines 227-229: Do not repeat m/z. The m/z is calculated for a given molecular formula not the opposite. So, “HRESI+TOF–MS (m/z, % rel. int.): 465.1022 [M+H]+ (100%), calculated for m/z 465.1033 [C21H20O12 + H]+, m/z 319.0441 [M–C6H11O5]+ (89%), calculated for myricetin at m/z 319.0454 [C15H11O8]+ ;” should be replaced by “HRESI+TOF–MS (m/z, % rel. int.): 465.1022 [M+H]+ (100%), calculated for C21H21O12: 465.1033; 319.0441 [M–C6H11O5]+ (89%), calculated for C15H11O8: 319.0454”.
The same for compound 2.
Section 3.2.
The discussion on the structural elucidation of the isolated compounds is very incipient. The authors have identified myricitrin and its constitutional isomer (compound 2) based on the mass and 1D and 2D NMR spectral data and in comparison with the spectral data of myricitrin previously reported. While the identification of myricitrin is, in this way, quite straightforward, the structural elucidation of compound 2 must be described in more detail. In particular, the authors must discuss the observed COSY and HMBC correlations and show how these lead to the proposed structure, once the structural difference between the two isomers is very subtle.
Also, the downfield deviation of the chemical shifts of carbons 4’ and 5’ of compound 2 relative to those of the same carbon atoms in compound 1 should be commented and explained.
Table 1.
The title is misleading as the NMR spectral assignment is for hydrogen and carbon atoms, not for compounds and should be changed.
The table is a bit confuse, particularly the header. It should be remade for clarity.
The table’s caption suffers from redundancy. Delete “key” and “isolated compounds (1 and 2), myricitrin (1), unifloratrin (2)”
Figure 1.
Should be split in two figures: one with the structures of the two identified compounds and another with the COSY and HMBC correlations for compound 2. These are unnecessary for myricitrin once it is not discussed.
Section 3.3.
In Section 2.7.2. it is said that, on the MTT test, cells treated with 10 % (v/v) DMSO (10 μL/well) served as additional controls. However, the authors do not make any reference in the discussion of cytotoxicity to the innocuity (or not) of DMSO on cell viability. Please comment.
Line 268, 274: The cytotoxicity values should be referred as CC50 (e.g. “ with CC50 values of 28.2±3.1 and 25.6±3.4 μg/100 μL”).
Lines 301-306: The authors justify the cytotoxicity of myricitrin by the presence of hydroxyl groups at specific positions, based on a previous report on the role of hydroxyl groups in the B-ring of flavonoids, namely quercetin and morin, in binding to triple helical RNA. The justification made in the paragraph is not conveniently explained and should be more elaborated for clarity. Moreover, given the structural similarity between compounds 1 and 2, a tentative correlation between the cytotoxicity of compound 2 and its structure could be provided.
Table 2.
The tables’ caption is redundant and possibly somewhat uncorrect. Delete “key”. Replace “alphabets” for “letters”. What the authors want to mean with: “values with different alphabets in superscript are significant at P<0.05”? It is intended to say that “different letters mean significant differences (p<0.05)”? “5-fluorouracil (5-FU), standard drug); isolated compounds (1 and 2); myricitrin (1); unifloratrin (2); non-tumorigenic human embryonic cell (HEK-293)” can be deleted. Definition of FU, compounds 1 and 2 and HEK-293 was previously provided.
Figure 2.
Caption should be reduced, for example, to “Cytotoxic activity of compounds isolated from Eugenia uniflora leaves”, once the heading of the figures is self-explanatory.
Section 3.4
Line 317: Please clarify the meaning of “purification”.
Table 3
What the authors want to mean with: “values with different alphabets in superscript are significant at P<0.05”? It is intended to say that “different letters mean significant differences (p<0.05)”?
Author Response
RESPONSES TO REVIEWER 2
General
The manuscript presents several minor typos that require correction. Some examples:
Line 33: Eugenia uniflora L must have a full stop after L.
Line 51: trans and cis must be italicized.
Line 102: n-hexane, the n must be italicized.
Line 169: 0.1 Mm.
Line 322, H2O2. The 2 must be subscript.
Line 324: “showed that 1”, 1 is not in bold.
Line334: in vivo must be italicized.
Line 346: in vitro must be italicized.
Once HEK-293, Hep-G2 and HeLa cell lines have been defined in the abstract, there is no need to repeatedly define the acronym of the cell lines throughout the manuscript.
The same for other acronyms such as MTT.
Response:
This has been revised to read “Eugenia uniflora L.
trans and cis are now italicized.
The n in n-hexane has been written in italics.
It has been corrected to read “0.1 mM”.
The ‘2’ in H2O2 has been revised as a subscript.
1 is now written in bold.
in vivo is now written in italics. In vitro has been italicized as well.
These corrections have been addressed in the revised manuscript.
Abstract
Line 13: According to the abstract the authors performed an “activity-guided fractionation of the plant”. This is not truly the case. In fact, the authors performed the fractionation of the EtOH-distilled water 97 (4:1) extract of the plant leaves. Please correct.
Response:
The abstract has been revised to read “activity-guided fractionation of the leaf ethanolic extract”.
Line 67: The authors stated that flavonoids “are polyphenols synthesized by plants in response to microbial infections [20].” In fact flavonoids act as plant defence mechanisms against biotic and abiotic factors. Why the authors were reductive?
Response:
The statement has been revised to read:
They are polyphenols synthesized by plants to act as defence mechanism against biotic (microbial infections) and abiotic stress [20].
Results and discussion
General: m/z should be in italic. Conventionally, in deuterated solvents names the notation “d” is italicized as in DMSO-d6. In this case, “6” must be subscript.
Response:
m/z has been italicized throughout the revised manuscript.
The d and 6 in DMSO-d6 have been written in italics and as subscript respectively.
Section 2.4.
Line 107: A “mixture” of n-hexane-EtOAc (100:0) is not a mixture, but a single solvent. A proportion is different from a percentage. Accordingly, hange “n-hexane-EtOAc (100:0, 90:10, 80:20, 70:30, 60:40, 50:50, 40:60, 30:70, 20:80 and 10:90%)” to “n-hexane, n-hexane-EtOAc (90:10, 80:20, 70:30, 60:40, 50:50, 40:60, 30:70, 20:80 and 10:90)”.
The same for the EtOAc-MeOH and DCM-MeOHmixtures.
Response:
This has been revised accordingly.
Section 2.7.2.
Line 165: Define CC50.
Line 185: Define IC50.
Response:
CC50 and IC50 have been defined in the revised manuscript.
Section 3
The authors say they performed an activity-guided fractionation of the EtOH-water extract of the plant leaves. However, there is no information on which activity served as the criteria to choose the EtOAc fraction to be further separated. Was it the cytotoxic activity, the antioxidant activity, both? The authors should clarify this.
In the same way, which was the criteria to choose subfraction F5 to be further fractionated by column chromatography? It is important to have the rational described.
Response:
EtOAc fraction was chosen because it gave the best cytotoxicity and antioxidant activity among the partition fractions as presented in Tables 2 and 3 respectively.
The subfraction F5 that was further fractionated was selected based on a rapid TLC bioautography screening with DPPH spray, which showed strong radical scavenging property. This has been addressed (Section 2.4).
Section 3.1.
Although both compounds 1 and 2 have 20 hydrogen atoms, the 1H NMR spectrum of compound 1 only records 11 hydrogen atoms, while that of compound 2 only records 12 hydrogen atoms. All hydrogen atoms must be attributed to the corresponding signals in the spectrum.
Line 225: Once compound 1 was identified and the molecular formula is given below “3.1.1. Compound 1 (myricitrin, C21H20O12)” should be replaced by “Myricitrin”.
Line 226: The yield provided is not a true yield as the plant’s content of myricitrin is not known. Therefore, it is misleading and should be removed. The same for compound 2.
Lines 227-229: Do not repeat m/z. The m/z is calculated for a given molecular formula not the opposite. So, “HRESI+TOF–MS (m/z, % rel. int.): 465.1022 [M+H]+ (100%), calculated for m/z 465.1033 [C21H20O12 + H]+, m/z 319.0441 [M–C6H11O5]+ (89%), calculated for myricetin at m/z 319.0454 [C15H11O8]+ ;” should be replaced by “HRESI+TOF–MS (m/z, % rel. int.): 465.1022 [M+H]+ (100%), calculated for C21H21O12: 465.1033; 319.0441 [M–C6H11O5]+ (89%), calculated for C15H11O8: 319.0454”.
The same for compound 2.
Response:
Kindly note the remaining “eight unassigned hydrogen atoms” exist in the compounds as hydroxyl groups. Three of these exist on the sugar as hydroxylated methine (CH) to affect its proton (CH-OH) shifts, three exist on quaternary carbon of the flavonoid ring B, while the remaining two hydroxyl group also exist on the quaternary carbons of the flavonoid ring C. However, the low proton signal at 12.65 ppm is due to an intermolecular hydrogen bonding between the proton of 5-OH (hydroxyl group at position C-5) and the oxygen of the neighbouring carbonyl group at position C-4.
After Section 3.1.1, compound 1 is now referred to as myricitrin.
After Section 3.2, which described the elucidation of compound 2, it is now referred to as unifloratrin.
The percentage yields of the isolated compounds have been removed. The authors only retain the weights obtained i.e., 29 mg and 52 mg respectively.
The MS data for 1 and 2 have been revised accordingly.
The discussion on the structural elucidation of the isolated compounds is very incipient. The authors have identified myricitrin and its constitutional isomer (compound 2) based on the mass and 1D and 2D NMR spectral data and in comparison with the spectral data of myricitrin previously reported. While the identification of myricitrin is, in this way, quite straightforward, the structural elucidation of compound 2 must be described in more detail. In particular, the authors must discuss the observed COSY and HMBC correlations and show how these lead to the proposed structure, once the structural difference between the two isomers is very subtle.
Also, the downfield deviation of the chemical shifts of carbons 4’ and 5’ of compound 2 relative to those of the same carbon atoms in compound 1 should be commented and explained.
Response:
The structure of the isolated compounds have been described, with more attention paid to the isomer of myricitrin. The HMBC experiment was used in particular to highlight the slight structural difference of 2.
Also, the characteristic downfield deviation of the chemical shifts of carbons 4’ and 5’ of compound 2 has been explained.
Table 1.
The title is misleading as the NMR spectral assignment is for hydrogen and carbon atoms, not for compounds and should be changed.
The table is a bit confuse, particularly the header. It should be remade for clarity.
The table’s caption suffers from redundancy. Delete “key” and “isolated compounds (1 and 2), myricitrin (1), unifloratrin (2)”.
Response:
Table 1 title has been revised accordingly.
The header has been remade and revised for clarity.
The “key” and “isolated compounds (1 and 2), myricitrin (1), unifloratrin (2)” have been deleted.
Figure 1.
Should be split in two figures: one with the structures of the two identified compounds and another with the COSY and HMBC correlations for compound 2. These are unnecessary for myricitrin once it is not discussed.
Response:
Figure 1 has been split into two figures.
Now, Figure 1 represents the structures of the two compounds, while the COSY and HMBC correlations for unifloratrin is presented in Figure 2.
Section 3.3.
In Section 2.7.2. it is said that, on the MTT test, cells treated with 10 % (v/v) DMSO (10 μL/well) served as additional controls. However, the authors do not make any reference in the discussion of cytotoxicity to the innocuity (or not) of DMSO on cell viability. Please comment.
Line 268, 274: The cytotoxicity values should be referred as CC50 (e.g. “ with CC50 values of 28.2±3.1 and 25.6±3.4 μg/100 μL”).
Response:
It was observed in the MTT test, that the 10% (v/v) DMSO used as an additional control demonstrated ≥99.9% cell viability, which suggests that the vehicle (dissolution solvent) did not in any way enhance the cytotoxicity of the extracts, fractions, and isolated compounds.
This has now been expressly stated in the discussion (section 3.3).
The cytotoxicity values are now mostly referred as CC50 values.
Lines 301-306: The authors justify the cytotoxicity of myricitrin by the presence of hydroxyl groups at specific positions, based on a previous report on the role of hydroxyl groups in the B-ring of flavonoids, namely quercetin and morin, in binding to triple helical RNA. The justification made in the paragraph is not conveniently explained and should be more elaborated for clarity. Moreover, given the structural similarity between compounds 1 and 2, a tentative correlation between the cytotoxicity of compound 2 and its structure could be provided.
Response:
This has been elaborated in a more specific terms by making reference to the structure-activity-relationship report, which highlighted the importance of C-3 and C-4 or C-4 and C-5 on bioactivity. These positions are known to enhance the distribution of electron cloud around the phenyl ring of flavonoids, which in turn makes them more liable to donate protons to form hydrogen bonds with the cell active site (conjugation), to inhibit oxidative and/or biological activities.
Table 2.
The tables’ caption is redundant and possibly somewhat uncorrect. Delete “key”. Replace “alphabets” for “letters”. What the authors want to mean with: “values with different alphabets in superscript are significant at P<0.05”? It is intended to say that “different letters mean significant differences (p<0.05)”? “5-fluorouracil (5-FU), standard drug); isolated compounds (1 and 2); myricitrin (1); unifloratrin (2); non-tumorigenic human embryonic cell (HEK-293)” can be deleted. Definition of FU, compounds 1 and 2 and HEK-293 was previously provided.
Response:
This has been revised accordingly.
Figure 2.
Caption should be reduced, for example, to “Cytotoxic activity of compounds isolated from Eugenia uniflora leaves”, once the heading of the figures is self-explanatory.
Response:
The caption has been reduced accordingly.
Section 3.4
Line 317: Please clarify the meaning of “purification”.
Response:
The word “purification” has been replaced with “fractionation”. The major highlight here is that there was considerable improvement in the antioxidant activity based on the fractionation work leading to isolation of the compounds.
This has been revised accordingly.
Table 3
What the authors want to mean with: “values with different alphabets in superscript are significant at P<0.05”? It is intended to say that “different letters mean significant differences (p<0.05)”?
Response:
This has been revised to read “different letters mean significant differences (P<0.05).
Thank you.
Round 2
Reviewer 2 Report
The authors have addressed properly the majority of the issues raised in the previous review. However, minor changes are still needed. Please see below.
General
Although the authors have chosen not to use the systematic IUPAC name, “5,7-dihydroxy-3-(3,4,5-trihydroxy-6-methyl-tetrahydro-pyran-2-yloxy)-2-(2,4,5-trihydroxy-phenyl)-chromen-4-one” should be better written as “5,7-dihydroxy-3-(3,4,5-trihydroxy-6-methyltetrahydropyran-2-yloxy)-2-(2,4,5-trihydroxyphenyl)chromen-4-one” to avoid unnecessary dashes (in IUPAC’s rules dashes are used only between letters and numbers).
It makes sense that compounds myricitrin (1) and 5,7-dihydroxy-3-(3,4,5-trihydroxy-6-methyltetrahydropyran-2-yloxy)-2-(2,4,5-trihydroxy-phenyl)chromen-4-one, or unifloratrin (2) are also identified by numbers (1 and 2). However, there must be coherence throughout the text: use the compound’s name or the compound’s name followed by the describing number between parentheses or only the number. These numbers must be in bold.
Introduction
Line 67: Biotic stresses can be of different types. Therefore, it makes no sense to refer microbial infections in particular.
Section 2.2.
Line 97: Change “The leaves were air-dried by protection from direct sunlight in a screen house” to “The leaves were air-dried under protection from direct sunlight in a screen house”.
Line 100: The brand is unnecessary. Change “Ziploc bag” by, for example, “plastic bag”.
Line 102: Use the more common form “80% EtOH”.
Lines 105 and 112: It is not the desiccator that is activated! Change to “kept inside a desiccator under….” and mention the dehydrating agent used (e.g. silica, anhydrous phosphorous pentoxide, etc.).
In the sentence “The concentrated extract (11.8% yield) was kept inside … afforded the leaf EtOH extract (11.8 % yield).” remove the yields. The total mass of this extract is given in the next sentence.
Line 110: The meaning of “The completeness of each partition fraction was monitored by TLC analysis.” is not clear. Rewrite for clarity.
Line 115: “The most antioxidant active EtOAc fraction (17.5 g)”. There was only one EtOAc fraction. Rewrite for clarity.
Line 121: “The subfractions were screened for the best free radical scavenging fraction, by spraying the developed TLC plates with 4 mg/mL DPPH solution in methanol. ”Where the TLC plates incubated in the dark? For how long? How did the active components appeared in the TLC plate? Elaborate for clarity.
Line 131: “suggesting them to be glycosidic compounds, with melting ranges of 196-198°C and 197-199°C.” How did the authors have measured the melting points of the isolated compounds on a TLC plate after spraying with 10% sulfuric acid and heating. Please explain. Indicate the apparatus used for measuring the melting points.
Section 3.1.1.
Line 244: Once compound 1 was identified and the molecular formula is given below replace “3.1.1. Compound 1 (myricitrin, C21H20O12)” by “Myricitrin”.
Line 253: Change” 3.1.2. Compound 2 (isomer of myricitrin, C21H20O12) 5,7-Dihydroxy-3-(3,4,5-trihydroxy-6-methyl-tetrahydro-pyran-2-yloxy)-2-(2,4,5-tri hydroxy-phenyl)-chromen-4-one (2)” to “Unifloratrin”.
Section 3.2.
Line 286: change “The molecular structures of 2…” to “The molecular structure of 2…”.
Table 1.
The table is still a bit confuse, particularly the header. DEPT data should be under Unifloratrin. There should be a header for Myricitrin comprising the acquired data and the literature data.
Section 3.3.
Line 314: “dissolution solvent (vehicle)” is redundant. Change to “solvent”.
Section 3.4
Line 237: “According to the result presented in Table 3, there was a considerable improvement in the antioxidant activity of E. uniflora leaves when the extract was fractionation into the various partition and chromatographic fractions, leading to the isolated compounds” Please rewrite for clarity.
Minor editing of English language required.
Author Response
RESPONSES TO REVIEWER'S COMMENTS
General
Although the authors have chosen not to use the systematic IUPAC name, “5,7-dihydroxy-3-(3,4,5-trihydroxy-6-methyl-tetrahydro-pyran-2-yloxy)-2-(2,4,5-trihydroxy-phenyl)-chromen-4-one” should be better written as “5,7-dihydroxy-3-(3,4,5-trihydroxy-6-methyltetrahydropyran-2-yloxy)-2-(2,4,5-trihydroxyphenyl)chromen-4-one” to avoid unnecessary dashes (in IUPAC’s rules dashes are used only between letters and numbers).
Response:
The IUPAC name has been corrected by removing unnecessary dashes.
It makes sense that compounds myricitrin (1) and 5,7-dihydroxy-3-(3,4,5-trihydroxy-6-methyltetrahydropyran-2-yloxy)-2-(2,4,5-trihydroxy-phenyl)chromen-4-one, or unifloratrin (2) are also identified by numbers (1 and 2). However, there must be coherence throughout the text: use the compound’s name or the compound’s name followed by the describing number between parentheses or only the number. These numbers must be in bold.
Response:
The compounds are now consistently described as myricitrin (1) and unifloratrin (2) in the revised manuscript.
Introduction
Line 67: Biotic stresses can be of different types. Therefore, it makes no sense to refer microbial infections in particular.
Response:
Microbial infections have been expunged from this statement.
Section 2.2.
Line 97: Change “The leaves were air-dried by protection from direct sunlight in a screen house” to “The leaves were air-dried under protection from direct sunlight in a screen house”.
Response:
The statement has been changed to "The leaves were air-dried under protection from direct sunlight in a screen house”.
Line 100: The brand is unnecessary. Change “Ziploc bag” by, for example, “plastic bag”.
Response:
The brand name "Ziploc bag" has been replaced with "plastic bag".
Line 102: Use the more common form “80% EtOH”.
Response:
The more common form, 80% EtOH, is now used in the revised manuscript.
Lines 105 and 112: It is not the desiccator that is activated! Change to “kept inside a desiccator under….” and mention the dehydrating agent used (e.g. silica, anhydrous phosphorous pentoxide, etc.).
Response:
The statement has been revised as "The concentrated extract was kept inside a desiccator under silica for 72 h to remove water from the organic extract; thus, afforded the leaf EtOH extract".
In the sentence “The concentrated extract (11.8% yield) was kept inside … afforded the leaf EtOH extract (11.8 % yield).” remove the yields. The total mass of this extract is given in the next sentence.
Response: The yields have been removed.
Line 110: The meaning of “The completeness of each partition fraction was monitored by TLC analysis.” is not clear. Rewrite for clarity.
Response:
The word "completeness" was introduced by one of the reviewers. However, the statement was revisited and has now been revised as follows:
The extract (114.3 g) was suspended in 417 mL of distilled water and successively partitioned with organic solvents in increasing order of polarity to afford n-hexane (2x500 mL, 8.1% yield), dichloromethane (DCM, 3x500 mL, 22.1% yield), ethyl acetate (EtOAc, 4x500 mL, 17.9% yield) and aqueous (45.7% yield) fractions. The solvent-partitioning process was TLC-monitored to ensure an exhaustive transfer of the organic fractions from the aqueous phase.
Line 115: “The most antioxidant active EtOAc fraction (17.5 g)”. There was only one EtOAc fraction. Rewrite for clarity.
Response:
The statement has been revised to read: "The EtOAc fraction (17.5 g) was further fractionated because it demonstrated the best antioxidant activity among the partition fractions".
Line 121: “The subfractions were screened for the best free radical scavenging fraction, by spraying the developed TLC plates with 4 mg/mL DPPH solution in methanol. ”Where the TLC plates incubated in the dark? For how long? How did the active components appeared in the TLC plate? Elaborate for clarity.
Response:
This has been revised as: "Subfraction F5 gave the strongest bleaching reaction, with yellow spots against a purple DPPH background after 5 min of incubation in the dark".
Line 131: “suggesting them to be glycosidic compounds, with melting ranges of 196-198°C and 197-199°C.” How did the authors have measured the melting points of the isolated compounds on a TLC plate after spraying with 10% sulfuric acid and heating. Please explain. Indicate the apparatus used for measuring the melting points.
Response:
The melting point determination of the compounds has been separated from the TLC analysis in the revised manuscript.
Section 3.1.1.
Line 244: Once compound 1 was identified and the molecular formula is given below replace “3.1.1. Compound 1 (myricitrin, C21H20O12)” by “Myricitrin”.
Line 253: Change” 3.1.2. Compound 2 (isomer of myricitrin, C21H20O12) 5,7-Dihydroxy-3-(3,4,5-trihydroxy-6-methyl-tetrahydro-pyran-2-yloxy)-2-(2,4,5-tri hydroxy-phenyl)-chromen-4-one (2)” to “Unifloratrin”.
Response:
The compounds are now presented as myricitrin (1), C21H20O12 and Unifloratrin (2), C21H20O12.
Section 3.2.
Line 286: change “The molecular structures of 2…” to “The molecular structure of 2…”.
Response: It has been corrected as molecular structure.
Table 1.
The table is still a bit confuse, particularly the header. DEPT data should be under Unifloratrin. There should be a header for Myricitrin comprising the acquired data and the literature data.
Response:
The DEPT is now under Unifloratrin.
There is now a header for myricitrin comprising the acquired data and the literature data.
Section 3.3.
Line 314: “dissolution solvent (vehicle)” is redundant. Change to “solvent”.
Response: The statement has been revised as solvent.
Section 3.4
Line 237: “According to the result presented in Table 3, there was a considerable improvement in the antioxidant activity of E. uniflora leaves when the extract was fractionation into the various partition and chromatographic fractions, leading to the isolated compounds” Please rewrite for clarity.
Response: This statement has been revised to read: "the ethyl acetate fraction of E. uniflora was more active than the mother extract, while each of the characterized compounds exhibited more antioxidant activity than the mother EtOAc fraction"
Comments on the Quality of English Language
Minor editing of English language required
Response: The revised manuscript has been checked for grammar, and spellings and revised accordingly.